# Influence of Implant Macro-Design, -Length, and -Diameter on Primary Implant Stability Depending on Different Bone Qualities Using Standard Drilling Protocols—An In Vitro Analysis

**DOI:** 10.3390/jfb14090469

**Published:** 2023-09-12

**Authors:** Milan Stoilov, Ramin Shafaghi, Helmut Stark, Michael Marder, Dominik Kraus, Norbert Enkling

**Affiliations:** 1Department of Prosthodontics, Preclinical Education and Dental Materials Science, Bonn University, 53111 Bonn, Germany; stoilov@uni-bonn.de (M.S.); helmut.stark@uni-bonn.de (H.S.); michael.marder@ukbonn.de (M.M.);; 2Department of Reconstructive Dentistry and Gerodontology, Bern University, 3012 Bern, Switzerland; dr.shafaghi@yahoo.com

**Keywords:** immediate loading, primary stability, insertion torque, ISQ, RFA, bone quality

## Abstract

(1) Background: Primary implant stability is vital for successful implant therapy. This study explores the influence of implant shape, length, and diameter on primary stability in different bone qualities. (2) Methods: Three implant systems (two parallel-walled and one tapered) with various lengths and diameters were inserted into polyurethane foam blocks of different densities (35, 25, 15, and 10 PCF) using standard drilling protocols. Primary stability was assessed through insertion torque (IT) and resonance frequency analysis (RFA). Optimal ranges were defined for IT (25 to 50 Ncm) and RFA (ISQ 60 to 80). A comparison of implant groups was conducted to determine adherence to the optimal ranges. (3) Results: Implant macro-design, -length, and -diameter and bone block density significantly influenced IT and RFA. Optimal IT was observed in 8/40 and 9/40 groups for the parallel-walled implants, while the tapered implant achieved optimal IT in 13/40 groups (within a 25–50 Ncm range). Implant diameter strongly impacted primary stability, with sufficient stability achieved in only one-third of cases despite the tapered implant’s superiority. (4) Conclusions: The findings highlight the need to adapt the drilling protocol based on diverse bone qualities in clinical practice. Further investigations should explore the impact of these adapted protocols on implant outcomes.

## 1. Introduction

The restoration of partially or completely edentulous patients with implants has emerged as a highly reliable and predictable treatment option with impressive survival and success rates [1]. To address the growing demand for reducing the waiting period before prosthetic rehabilitation of implants and to gain further insights into osseointegration events, an immediate loading protocol was developed [2,3,4]. In delayed loading, the loading occurs two months after implant placement, whereas in immediate loading, it takes place within one week of implant placement. Ensuring optimal primary implant stability is a crucial prerequisite for successful immediate loading and long-term success [5,6,7,8]. In particular, the rehabilitation of fully edentulous patients by using full-arch fixed prostheses with several implants has been further developed by applying immediate-loading protocols with the connection of the prosthesis on the same day of the surgery and is considered a predictable procedure [9,10,11]. In the case of full-arch restorations, lower rates of primary stability have also been reported to be successful [12,13]. Therefore, for immediate loading, achieving a primarily quadrangular interlocking of implants appears to be beneficial for the survival prognosis [14].

Primary stability refers to the lack of movement of the implant shortly after insertion into the bone site or the biomechanical interaction between the implant surface and the peri-implant bone during implant placement [15,16,17]. It is important to distinguish primary stability from secondary stability, which results from the healing process itself [18]. Optimal primary implant stability is believed to promote bone cell differentiation [5]. On the other hand, reduced primary stability may lead to micro-movements, fibrous tissue formation, and early implant failure [19,20,21]. Primary stability is predominantly influenced by the implant design, the surface treatment, the specific properties of the implant site, and the surgical technique used for preparation [22,23]. In addition, implant diameter and length are positively correlated with primary implant stability [24].

Various methods have been described for the quantitative assessment of primary implant stability, with two commonly cited approaches being implant insertion torque (IT) and resonance frequency analysis (RFA). Resonance frequency analysis is expressed as the implant stability quotient (ISQ) and has been extensively used in the literature [25,26,27]. In the clinical setting, ISQ measurements are employed as indirect indicators to determine the appropriate timing for effective implant loading and as predictive markers for potential implant failure [28]. RFA involves evaluating how an implant-attached piezo-ceramic element responds to a vibration stimulus comprised of small sinusoidal signals, ranging from 5 to 15 kHz, with incremental steps of 25 Hz. The highest amplitude of this response is then translated into a parameter known as the implant stability quotient (ISQ), which falls within the range of 0 to 100 [29]. The ISQ value positively reflects the overall mechanical stability of the implant–bone connection [28].

Insertion torque is an easily obtainable and representative parameter for estimating the primary implant stability during implant placement [30]. IT can be measured using standardized hand ratchets or special surgical motors. It reflects the cutting resistance of the bone during implant placement and is measured in Newton centimeters (Ncm) [31]. Increased insertion torque helps achieve primary stability by reducing implant micro-motion. Today, both IT and RFA are the most important values for clinically determining implant primary stability. Several clinical studies have used the two aforementioned stability measurement methods to try to establish a predictive value for the probability of osseointegration, occasionally in highly complex protocols, such as immediate loading [32,33].

An ISQ of 60 and an insertion torque of 35 Ncm are generally considered favorable conditions for optimal primary implant stability [34].

However, the ideal implant insertion torque for immediate loading remains a topic of lively debate. Weigl and Strangio [35] suggest that implant survival rates do not differ significantly between 25 Ncm and 32 Ncm. Higher torque rates, up to 40 or 50 Ncm, appear to be associated with better implant survival rates [36]. To ensure adequate primary stability, especially for immediate loading, some researchers recommend an insertion torque of at least ≥32 Ncm or above 35 Ncm [37]. Animal research indicates a strong association between an insertion torque of ≥32 Ncm and successful implant outcomes [38]. In a systematic review and meta-analysis, Benic et al. concluded that immediately and conventionally loaded single-implant crowns are equally successful regarding implant survival and marginal bone loss. This conclusion was primarily derived from studies evaluating implants inserted with a torque ≥ 20 to 45 Ncm or ISQ values ≥ 60 to 65 and with no need for simultaneous bone augmentation [39]. Furthermore, a randomized controlled clinical study by Degidi et al. [40] on immediate implant placement and immediate restoration (inclusion criteria: IT > 25 Ncm and ISQ > 60) showed that all implants were osseointegrated and clinically stable at the follow-up examinations. Although Greenstein and Cavallaro [30] stated in a literature review that IT rates higher than 50 Ncm did not appear to damage bone, other research showed that increasing the insertion torque beyond a certain level may not be suitable for every implant system or bone type [41,42]. It could lead to higher peri-implant bone remodeling and buccal soft tissue recession than implants inserted with a regular IT (<50 Ncm) [41]. In addition, Marconcini et al. showed that implants placed with a higher insertion torque (>50 Ncm) in the mandible led to greater bone resorption and mucosal recession than those registered for implants placed with a regular IT (<50 Ncm) [42]. Taken together, in our hands, an IT of more than 25 Ncm and below 50 Ncm and an ISQ > 60 seem to be reliable clinical values for primary implant stability to be successful for immediate loading protocols.

It is crucial to consider various factors in the clinical situation, especially for immediate loading protocols, to achieve adequate primary stability. These factors include implant geometry, implant length and diameter, and a proper drilling protocol based on bone quality [43]. In areas with predominantly cortical and hard bone tissues, slow drilling and tapping are suggested to avoid thermal and mechanical necrosis [44]. Conversely, in areas with low bone quality, osteotomes, undersized drilling, and implant insertion without tapping have been recommended to increase primary stability [45,46,47,48].

Bone density plays a significant role in the choice and application of the implant system. In soft cancellous bone, using implants with increased length and apically located screw threads has shown positive effects on primary stability [49]. Implants with deeper thread depth also appear to increase primary stability in areas of poor bone quality [50]. However, Makary et al. [51] suggested that this effect may be beneficial only in soft bone (D3 and D4).

Regarding implant geometry, parallel-walled and tapered implants are mainly used in clinical practice. Tapered implants often show higher insertion torque values than parallel-walled implants, though no significant differences in implant failure rate between the two geometries have been reported [5]. Due to higher primary implant stability, tapered implants are therefore more likely to be used for immediate loading protocols [52].

The aim of the present study was to develop clinical recommendations for implant selection to achieve optimal primary implant stability based on bone density and quality. Several other studies also addressed this issue. However, these investigations frequently involved the utilization of diverse implant types sourced from different manufacturers, each subjected to varying implant surface treatment procedures [52,53]. In addition, different drill sets had to be used [53,54]. However, in the context of practical clinical applications, the assessment of bone quality primarily occurs during the drilling procedure and is contingent upon the surgeon’s expertise [55]. This study seeks to compare the impact of different implant types and macro-geometries on primary stability under the condition of uniform drill set utilization for implant bed preparation. To achieve this goal, a standardized in vitro model was used, employing polyurethane blocks with varying densities. The focus was on evaluating primary implant stability across three distinct implant types and associated macro-geometries, all subjected to the same surface treatment. Furthermore, different implant lengths and diameters were investigated, utilizing an identical drilling protocol. The quantification of primary stability was determined by both insertion torque (IT) and resonance frequency analysis (RFA) and evaluated in accordance with a predefined range for successful immediate loading protocols (IT: 25–50 Ncm; ISQ: ≥60). The first null hypothesis assumed that there are no differences between the different implant types in different bone qualities under a standard drilling protocol related to the optimal primary stability range. To gain further insight into implant behavior, we tested another null hypothesis that postulated no differences in the effects of implant dimensions (length and diameter) on primary stability at different bone densities and implant types.

The final null hypothesis assumed that the insertion torque is solely determined by the contact surface area with the bone, irrespective of the implant’s macro-geometry and -dimensions.

## 2. Materials and Methods

In the study, three different implant types with different lengths and diameters (all from SIC invent AG, Basel, Switzerland; Table 1) were placed in solid rigid polyurethane foam blocks (Sawbones, Vashon, USA) with four different densities (10 PCF, 15 PCF, 25 PCF, and 35 PCF; PCF = pounds per cubic). For primary implant stability, insertion torque (IT) and resonance frequency analysis (RFA) were recorded.

### 2.1. Implant Characteristics

All utilized implants underwent a conditioning process involving a blasting procedure using zirconia beads and subsequent acid cleaning (SICmatrix^®^). This process resulted in a moderate surface roughness (Sa = 1.0 µm). The study evaluated the performance of three distinct dental implant macro-designs:-Group I: Cylindrical implant (SICace^®^)

The SICace^®^ cylinder screw implant has a cylindrical basic shape with an apical taper. The self-tapping wide trapezoidal thread design is equipped at the apex with pronounced chip spaces and an aggressive thread cut. SICace^®^ implant is recommended by the manufacturer for all indications in oral implantology in D1 to D3 bone quality.

-Group II: Cylindrical implant (SICmax^®^)

The SICmax^®^ cylinder screw implant has a two-stage cylindrical basic shape with an apical taper. The self-tapping, pointed trapezoidal thread design is broadly rounded at the apex and has relatively small chip spaces. In the crestal area, the implant has an enlarged thread core diameter and a double micro-thread. Due to the combination of a gentle apex without direct thread cutting, the SICmax^®^ implant is particularly suitable for soft bones and in the maxillary posterior region, especially for all forms of sinus floor elevation. The SICmax^®^ implant is especially recommended for all indications in D2 to D4 bone quality and for immediate implantation.

-Group III: Tapered implant (SICtapered^®^)

The SICtapered^®^ cylinder screw implant has a conical basic shape with a two-step cylindrical coronal area. The self-tapping, very pointed trapezoidal thread design is equipped at the apex with pronounced chip spaces and an aggressive thread cut. In the crestal area, the implant has an enlarged thread core diameter. The SICtapered^®^ implant is recommended for all indications in D1 to D4 bone quality and also for immediate implantation.

### 2.2. Implant Site Preparation

To prepare the implant site, a CAD drawing was created, which evenly transferred a matrix of rows and columns to the various polyurethane foam blocks. This matrix formed the basis for the drilling axes of the planned 180 implant cavities per polyurethane foam block for all implant types and sizes (n = 6 per group; a total of 720 implant site preparations for all four bone qualities and implant types; Figure 1). The drilling was carried out with a high-precision, CNC-controlled milling machine (Figure 2). The drills used were original SIC^®^ drills, and the drill sequence corresponded to the SIC^®^ drilling protocol (drilling speed: 300 rpm). This ensured that the drilling cavities corresponded exactly to the ideal contours of the SIC^®^ tools, without manual error influences, such as angle deviations or incorrect drilling depths. The different implant types, implant diameters, and lengths, as well as the drilling sequences (last drill and crestal drill), are given in Table 1.

### 2.3. Implant Insertion and Measurement of Implant Stability

All implants were inserted in the prepared implant sites using a torque-controlled surgical motor unit and a contra-angle handpiece (Implantmed Plus and WS-75L; W&H Deutschland GmbH, Laufen, Germany) at 25 rpm by one experienced implantologist. The used surgical motor unit enabled the recording of the insertion torque during the whole implant insertion process. The maximum insertion torque of the surgical motor unit is 80 Ncm. For the analysis of the insertion torque, the maximum insertion torque value for every single implant during the insertion was noticed and used for statistical analysis. If during the implant insertion the maximum torque value of 80 Ncm was reached and the implant shoulder did not finish flush with the polyurethane foam block (the implants could not be epicrestally inserted), the length of the protruding part of the implant was measured in millimeters and listed. In addition to the insertion torque, a resonance frequency analysis (RFA) was assessed by an electronic device (Osstell^®^ IDx, Osstell, Göteborg, Sweden) after mounting an appropriate SmartPeg (Type 92) on every implant. For every implant, the mean implant stability quotient (ISQ) value of three RFA assessments was calculated and listed.

### 2.4. Statistical Analysis

Mean maximum ITs ± standard deviations (SDs) and mean ISQs ± SDs were calculated, and the data were further processed by statistical analysis. The sample size was n = 6 for every sample (total N = 720). With regard to primary implant stability, an optimal range of 25 to 50 Ncm for IT and >60 for ISQ was defined. Critical ranges were defined for IT values below 25 Ncm and above 50 Ncm and for ISQ values below 60. For a comparison of the three different implant macro-designs, we descriptively demonstrated how frequently the values for IT and ISQ met the optimal ranges in different bone qualities.

To detect differences in the impact of implant length and diameter on IT across different bone qualities, a general linear model was fitted with implant length and implant diameter as covariates and block type and implant type as fixed factors, with log implant torque serving as the dependent variable. Samples with an insertion torque value of ≥80 Nm were excluded from the analysis, as this was the maximum torque for the surgical motor unit. Furthermore, implants of the largest dimensions (diameter: 5 mm; length: 14.5 mm) were excluded from the analysis, as in most cases they exceeded the maximum torque in harder blocks. If heteroscedasticity was present in the log data, estimates with heteroscedasticity-consistent standard errors (HC4) were calculated.

To test whether insertion torque can be adequately predicted by contact surface, a general linear model with block type and implant type as fixed factors and surface as a covariate was fitted with insertion torque as the dependent variable. The contact areas to the bone of all implants used were specified by the manufacturer and are listed in Table 2.

Significant interactions in both models were considered in the analysis, and simple effects were interpreted when necessary. Main effects were interpreted when permitted by the type of interaction. SPSS (Version 27, IBM, Armonk, NY, USA) was used for analysis. We assumed the linear model to be a sufficient approximation for extracting basic conclusions on the impact of implant dimensions on IT. For pairwise comparisons, *p*-values were adjusted with Bonferroni correction, and the results are noted as estimated values ([lower 95% CI, upper 95% CI]). Graphs were created with GraphPad Prism 6 (GraphPad Software, San Diego, CA, USA).

## 3. Results

Overall, implant macro-design, implant length, and diameter, as well as the density of the artificial bone blocks, had an influence on IT (Figure 3) and RFA (Figure 4). The implant length and diameter, as well as the simulated bone quality, were positively correlated with IT and RFA. In the 35 PCF blocks, IT values of 80 Ncm could be observed with implants, some of which could not be fully inserted. On the other hand, no IT of more than 25 Ncm or ISQ of >60 was recorded in any group with the standard drilling protocol in the softest polyurethane block (10 PCF). When comparing the three implant macro-designs, an optimal IT could be recorded for the two parallel-walled implants, SICace^®^ and SICmax^®^, in a total of 8/40 and 9/40 groups, respectively, while an IT of 25–50 Ncm could be achieved for the tapered implant in a total of 13/40 groups (Figure 3).

Considering the achieved ISQ values for all investigated implant types, lengths, and diameters, a value above 60 ISQ could be measured for SICace^®^ in 23 of 40 groups (Figure 4). For SICtapered^®^, a total of 24 of 40 groups showed ISQ values in the target range, while for SICmax^®^, 25 of 40 groups were within the target range. None of the tested implants achieved an ISQ value above 60 in the softest polyurethane block (10 PCF), whereas ISQ values above 60 were generally recorded in all implant groups in the 35 and 25 PCF artificial bone blocks. In the 15 PCF polyurethane block, only three SICace^®^ groups achieved an ISQ value in the target range, while five groups each showed ISQ values above 60 in the SICmax^®^ and SICtapered^®^ implants.

As expected, a strong effect of block density on the insertion torque was detected over all implant types (F(3, 596) = 1817, *p* < 0.001), with the mean insertion torque almost six times (IT: 5.6 [5.3, 5.8]) higher in PCF 35 when compared to PCF 10 (*p* < 0.001).

The general linear model revealed a positive relationship between implant dimensions and insertion torque for all implant types (F(1, 597) = 1069, *p* < 0.001, for diameter; F(1, 597) = 794, *p* < 0.001, for implant length). Overall, in the examined range, every additional mm in length resulted in an overall 10.1% [9.3%, 10.8%] increased insertion torque. Similarly, an increase in implant diameter of 1 mm resulted in an 85% [78%, 91%] higher insertion torque. However, besides being a positive predictor, the degree of association between implant dimensions and IT was heavily dependent on block density and implant type.

The analysis of the impact of implant type on IT in low bone quality (PCF 10 and PCF 15) revealed significant differences between the implants (F(2, 315) = 31.9, *p* < 0.001). SICta-pered^®^ yielded significantly better stability when compared to SICace^®^ (18.4% [12.5%, 24.8%]; *p* = 0.001) and slightly better stability when compared to SICmax^®^ (5.3% [0.2%, 10.9%]; *p* = 0.041), even though the effect of implant type was considerably smaller than that of implant length or diameter.

The model further revealed differences in the impact of implant length on IT between the SICace^®^, SICmax^®^, and SICtapered^®^ implants. Both SICmax^®^ and SICtapered^®^ exhibited significant systematic interactions between bone density and implant length (SICmax^®^: F(3, 185)= 25.2, *p* < 0.001; SICtapered^®^: F(3, 198) = 37.2, *p* < 0.001), meaning that implant length contributed less to insertion torque in soft blocks (PCF 10 and 15) when compared to hard blocks. This was also reflected in the effect sizes, as well as the regression coefficients (*b*) for implant length. The regression coefficient illustrates the degree of association between the predictor variable and the insertion torque (Table 3). It reflects the percentage increase in IT, when implant length/diameter is increased by 1 mm.

Such a straightforward relationship was not observed for SICace^®^, where the influence of implant length was only significantly higher in hard blocks (PCF 35) when compared to blocks of other qualities (F(3, 186) = 9.4, *p* < 0.001), and the effect sizes were not as pronounced as in the SICmax^®^ or SICtapered^®^ implants.

Whereas the impact of implant length on IT exhibited a significant change with block density, implant diameter did not systematically exhibit this relationship to a relevant ex-tent, despite the presence of significant differences (Table 3). The ratios of the regression coefficients for implant length and diameter thus suggest that diameter might prove a more reliable factor for implant stability in lower bone densities, at least for SICmax^®^ and SICtapered^®^ implants (Table 3).

To test the hypothesis that the effect of contact surface area is sufficient to predict insertion torque, we evaluated whether implant type still exhibits a relevant effect on IT when surface area is included in the model. We detected significant differences between the implant types depending on block density that could not be attributed to contact surface (smallest difference (PCF10): F(2, 158) = 6.5, *p* = 0.002, ηp^2^ = 7.6%). Furthermore, the model with surface as covariate exhibited a significant and visibly worse fit to the data when compared to the model that included implant length and diameter instead (*R*^2^_surface_: 0.893; *R*^2^_dimensions_: 0.961).

## 4. Discussion

In modern dental implantology, in addition to concepts for the long-term success of the implants (e.g., avoidance of peri-implantitis) [1,56], the focus is also on shortening the treatment time until the final restoration in order to increase the comfort, quality of life, and satisfaction of the patient [57,58,59]. Today, immediate implant placement and immediate loading protocols play an increasingly important role. In this case, the implant is placed in the empty extraction socket directly after the removal of a tooth, whereby the apical bone area or existing bone septa in particular are used to achieve sufficient primary stability. Today, a more oral placement of the implant in the alveolar socket is preferred [60], whereby the entire lumen of the alveolar socket is no longer filled with an implant, but a gap to the vestibular bone lamella usually remains [61,62]. After implant placement, this gap is filled primarily with bone substitute material that is difficult to resorb as resorption protection [63,64,65]. Primary wound closure can be achieved by using a temporary restoration (immediate loading) or by using an individualized healing abutment. This procedure allows the existing soft tissue to be stabilized and preserved with the aid of immediate implant placement if the clinical indication is adequate (e.g., given existing vestibular bone lamella, sufficient soft tissue supply, or no major inflammatory processes) [66]. This allows a significant reduction in treatment time for the patient and avoids uncomfortable provisional restorations. In addition to saving time, this can often reduce treatment costs. The removal of a tooth is always associated with a resorption of soft and hard tissue in the course of wound healing [67]. In the case of classic late implant placement (implants are placed only after complete osseous healing, usually after 6 months), this can lead to the need for soft and/or hard tissue augmentation in addition to implantation. In addition to the abovementioned prerequisites for immediate implant placement and immediate loading, the achievement of sufficient primary implant stability also plays a decisive role regarding success. For this purpose, implant manufacturers offer various macro-designs of implants, most of which feature a tapered implant geometry and/or progressive thread configurations [24]. In addition to the macro-design of implants, other factors also play an important role in the primary stability of implants: bone density, implant length and diameter, and the precision of implant bed preparation [52,68]. While bone density cannot be influenced in advance, the practitioner can influence primary stability through the choice of implant (macro-design, -length, and -diameter), as well as through the drilling protocol [24,69,70].

Therefore, the aim of our study was to evaluate to what extent implant primary stability in different bone qualities is influenced by different implant macro-designs, -lengths, and -diameters. For this purpose, a standardized in vitro model was used, using rigid polyurethane foam blocks of different densities to simulate different bone qualities. In clinical situations, many biological factors, such as bone response, individual characteristics, and local variations in human bone, also affect the primary stability. However, the American Society for Testing and Materials (ASTM F-1839-08) (“Standard specification for Rigid Polyurethane Foam for Use as a Standard Material for Test Orthopedic Devices for Instruments”) has recognized polyurethane foam blocks as alternative materials for biomechanical tests, even for dental implant evaluations [71]. These materials do not replicate the structure of human bone but have consistent mechanical properties that are similar to those of bone tissue. Furthermore, they are very reliable and easy to use, require no special handling, and are characterized by linear elastic and constitutive isotropic symmetry [72,73]. As has already been reported by Comuzzi et al. [74], polyurethane foam sheets are the most suitable materials for in vitro application to simulate the consistency and different densities of bone tissue and to compare dental implants and bone screws.

Various artificial bone block configurations are available. On the one hand, the density of the blocks can be varied to simulate different bone qualities; on the other hand, e.g., single-layer blocks can be distinguished from double-layer blocks. According to Misch, four different bone qualities are clinically distinguished, ranging from D1 (hardest bone) to D4 (softest bone) [75].

To simulate all possible clinical conditions, artificial bone blocks with four different densities (35, 25, 15, and 10 PCF) were used. All polyurethane foam blocks used were in the single-layer configuration.

Other studies that also address the issue of the primary stability of implants partly use double-layer configured foam blocks. This simulates a clinical situation that most closely corresponds to cases of late implant placement, where complete bone healing has taken place after tooth removal. Two layers of polyurethane foam are placed on top of each other, with the top layer usually being 1–2 mm thick and having a higher density than the bottom layer. This layering simulates the bone situation with an existing crestal cortical bone (hard bone) and an underlying cancellous bone (softer bone). The use of an additional cortical layer generally allows higher insertion torques and thus higher primary stability to be achieved than with the sole use of softer single-layer blocks [53,76].

In our study, we simulated the clinical situation of immediate implant placement. Thus, we decided to use monolayer blocks. Since, nowadays, a complete filling of the alveolar socket with an implant is not suitable, the insertion torque and thus the implant primary stability in these cases are determined less by the cortical bone portion and more by the bone structures located further apically. Particularly for the extremely soft bone, the results show that only low insertion torques can be achieved, which are all below the recommended insertion torques of at least 20 Ncm in the literature for the immediate loading of single implant crowns [39]. The polyurethane foam blocks used here with a density of 10 PCF simulate a very soft cancellous bone structure. In preliminary tests, implants could be inserted here without any implant bed preparation at all. The use of double-layer blocks with an additional cortical layer would probably have resulted in significantly higher insertion torque values. A study by Comuzzi et al. [76] showed that the placement of standard-length implants in bilayer polyurethane foam blocks almost doubled the insertion torque compared to the monolayer block configuration (IT: 20.9 Ncm vs. 11.3 Ncm). Here, the implants were placed in monolayer polyurethane foam blocks with 10 PCF and in blocks with an additional 1 mm cortical layer (30 PCF). These results show that immediate implant placement and immediate loading is not possible or very difficult to achieve with standard drilling protocols in extremely soft bone qualities. Under-preparation of the implant site, which is also recommended in the literature for such cases, could be helpful here [69,77]. However, the additional modification of the drilling protocols was not planned in the present study but should be considered in future studies. In any case, immediate implant placement with immediate loading is not recommended in cases with very soft bone quality. A two-stage approach should be followed here, in which closed implant healing should take place first and then late loading can be performed after complete osseointegration and uncovering of the implant.

In clinical practice, bone quality is mainly determined during the drilling procedure and depends on the surgeon’s experience [55].

Another way to determine bone quality prior to surgery would be to analyze Hounsfield units using cone-beam CT images [78].

However, this is the subject of controversial discussions in the literature [79], and it is not possible on the basis of classical two-dimensional X-ray images. As already mentioned, many implant manufacturers have developed various implant types for different indications. However, different drill sets often have to be used. With the implants used in this study, the implant site preparation for all three implant types can be performed with the same drill set. Transferred to the clinic, this means that the implant type can be changed during drilling depending on the bone quality.

The primary stability of implants depends not only on the type of implant or on biological factors, such as bone quality, but is also influenced by the precision of implant site preparation. In many studies, preparation is performed freehand, which leads to some variation in the placement and the quality of the drill hole and may thus introduce bias in the analysis of the data [80]. Therefore, all implant site preparations were performed with the help of a CNC milling machine using the original implant drills. Through this standardization, even small differences in the analysis of implant primary stability can be attributed solely to the different bone densities as well as implant types, lengths, and diameters.

To evaluate the performance of different implant types, lengths, and diameters in terms of primary stability in different bone qualities, we defined an adequate or optimal insertion torque range. This starts at 25 Ncm, which is considered as a minimum value for an immediate loading [35], and extents to a maximum value of 50 Ncm. This value should not be exceeded, as higher values may lead to higher peri-implant bone remodeling, buccal soft tissue recession, and greater bone resorption [41,42]. With the standard drilling protocol, the tapered implant type showed the best performance (13 of 40 groups lay within the appropriate range), followed by the two parallel-shaped implants (SICace^®^: 8 of 40 groups; SICmax^®^: 9 of 40 groups). Especially in the two soft bone blocks, sufficient primary stability could be achieved with the tapered implant at higher sizes. This is in line with several other studies that have also shown that tapered implants can generally achieve higher torques in soft bone compared to parallel-walled implants [50,54,66]. In contrast, none of the SICace^®^ implants (not even the largest available size (5.0 × 14.5 mm)) could achieve a sufficiently high insertion torque (max. 21 Ncm) for an immediate loading. The other parallel-walled implant, SICmax^®^, on the other hand, was able to achieve sufficient primary stability in some cases when large sizes were used. This also corresponds to the manufacturer’s recommendations for this implant. The SICmax^®^ implant was designed by the manufacturer for the maxilla in particular, also in conjunction with sinus floor elevation, and is intended to achieve higher torques in the softer maxillary bone. Thus, this implant is also recommended for immediate implantation and immediate loading, which has been in part proven by our data.

In addition to implant performance in soft bone, however, torque development in hard bone is also of great interest. In our setting, harder bone, long implants, and large-diameter implants cause an insertion torque that is far above the upper limit of 50 Ncm. Interestingly, the conical implants also showed the best performance in the harder polyurethane foam blocks. Thus, the tapered implant showed the best all-round performance across all bone qualities. Nevertheless, it has to be mentioned that in hard bone a much too high insertion torque (>50 Ncm) was achieved with the standard drilling protocol in most of the implant groups investigated. At the large sizes, especially, some implants could not even be inserted to their full length. Transferred to the clinical situation, this can lead to considerable pressure overload and necrosis in the peri-implant bone and ultimately to implant loss. Therefore, the drilling protocol should be adjusted for hard bone, longer implant lengths, and larger implant diameters. In such cases, the manufacturer recommends at least the use of a tap or an overextended drilling by the use of the next-larger drill. As mentioned for soft bone, future studies should also evaluate the adaptation of the drilling protocol.

Our study shows that implant primary stability is positively correlated with implant type, length, and diameter and negatively influenced by softer bone qualities. This is also in agreement with many other studies. The following question still arises for clinical practice: Which parameter (implant length vs. implant diameter) has a higher influence on primary stability, especially in soft bone conditions? Therefore, a linear model was performed with bone quality and implant type as fixed factors and implant dimensions as covariates.

In general, the influence of implant length on insertion torque decreased with decreasing bone quality, whereas the influence of implant diameter on insertion torque was not systematically affected, at least not in mid to low bone qualities. Furthermore, we could show that the implant surface in contact with the bone performs worse than implant length and implant diameter in predicting primary implant stability in our study. Thus, all postulated null hypotheses were rejected.

For the clinic, this means that with softer bone the implant diameter might play an increasingly important role in order to achieve an optimal insertion torque and primary stability.

## 5. Conclusions

Our study shows that implant primary stability is particularly dependent on bone quality and can be influenced by the selection of a suitable implant type, implant length, and diameter. However, with very soft bones, this influence is sparse. This results in the following recommendations for clinical practice:Tapered implants are particularly suitable for achieving higher torques in soft bone compared to standard parallel-walled implants. Therefore, tapered implants seem to be particularly suitable for immediate implantation and immediate loading.Larger implant sizes lead to higher insertion torques in soft bone and thus to higher primary stability. The increase in diameter seems to play a greater role than the implant length, so that wider implants are preferable in cases of soft bone and immediate loading.The development of the insertion torque must also be observed in the hard bone in order to avoid pressure overloads and necrosis in the peri-implant bone. Even with smaller implant sizes and the standard drilling protocol, critical values (e.g., >50 Ncm) can be exceeded.

Although the tapered implant showed the best all-round performance using the standard drilling protocol, adequate implant primary stability (25 to 50 Ncm) could only be achieved in one-third of the cases. Transferred to clinical practice, this means that very often an additional adaptation of the drilling protocol to the clinical situation, in particular the bone quality, is necessary. Therefore, further studies should additionally investigate the influence by adapting the drilling protocol (under-/over-preparation of the implant site) to bone qualities.

## Figures and Tables

**Figure 1 jfb-14-00469-f001:**
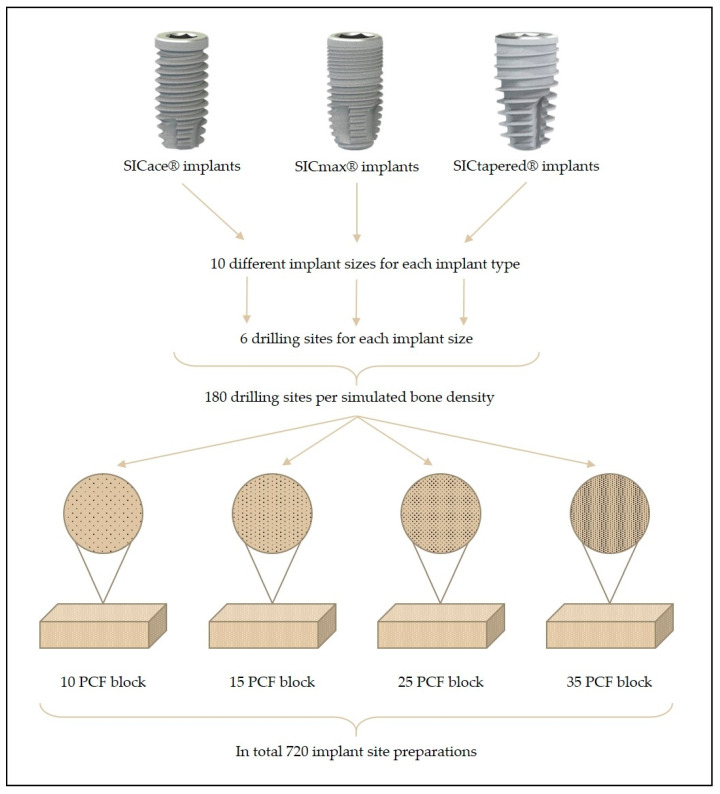
Schematic illustration of the implant site preparations and the study design.

**Figure 2 jfb-14-00469-f002:**
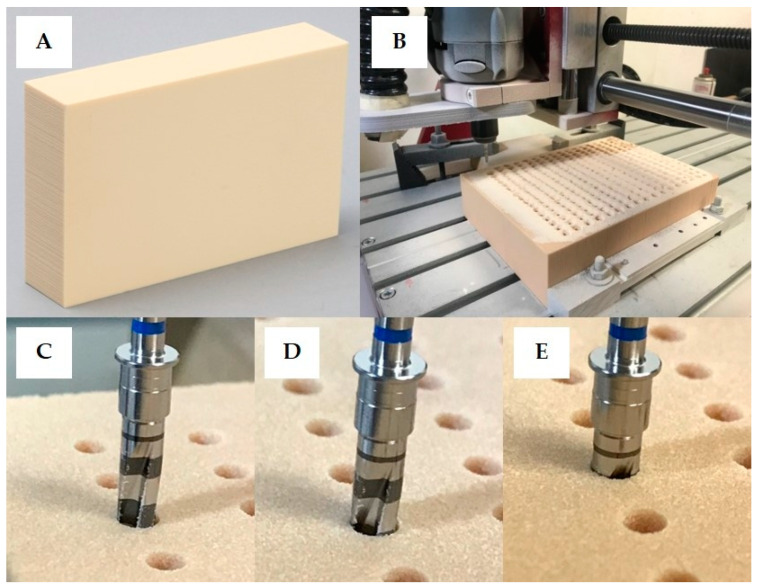
Experimental set-up: (**A**) polyurethane foam block (Sawbones, Vashon, WA, USA); (**B**) standardized implant site preparation using CNC milling machine; (**C**–**E**) standard drill (blue, 3.1 mm diameter) in various drilling depths (**C** = 7.5 mm; **D** = 9.5 mm; **E** = 13 mm).

**Figure 3 jfb-14-00469-f003:**
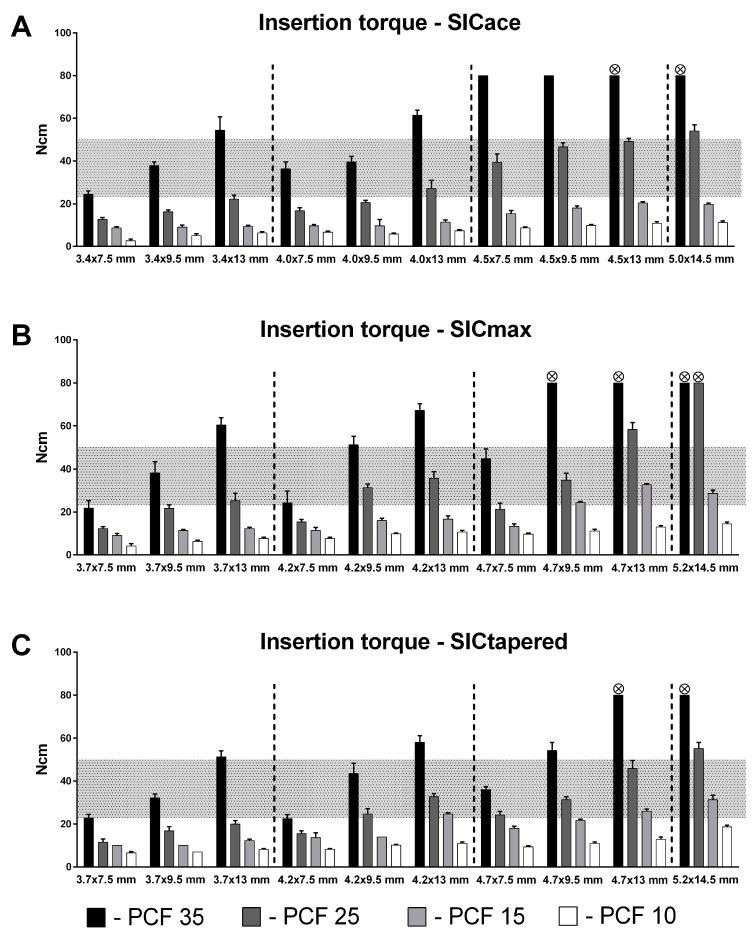
Insertion torque of SICace^®^ (**A**), SICmax^®^ (**B**), and SICtapered^®^ (**C**) implants with different lengths and diameters in four different polyurethane foam block densities (35, 25, 15, and 10 pounds per cubic (PCF)). The grey shaded area represents the optimum insertion torque range from 20 to 50 Ncm. Symbols at the top of the bars indicate that implants could not be inserted to their full length when the maximum torque of 80 Ncm was reached.

**Figure 4 jfb-14-00469-f004:**
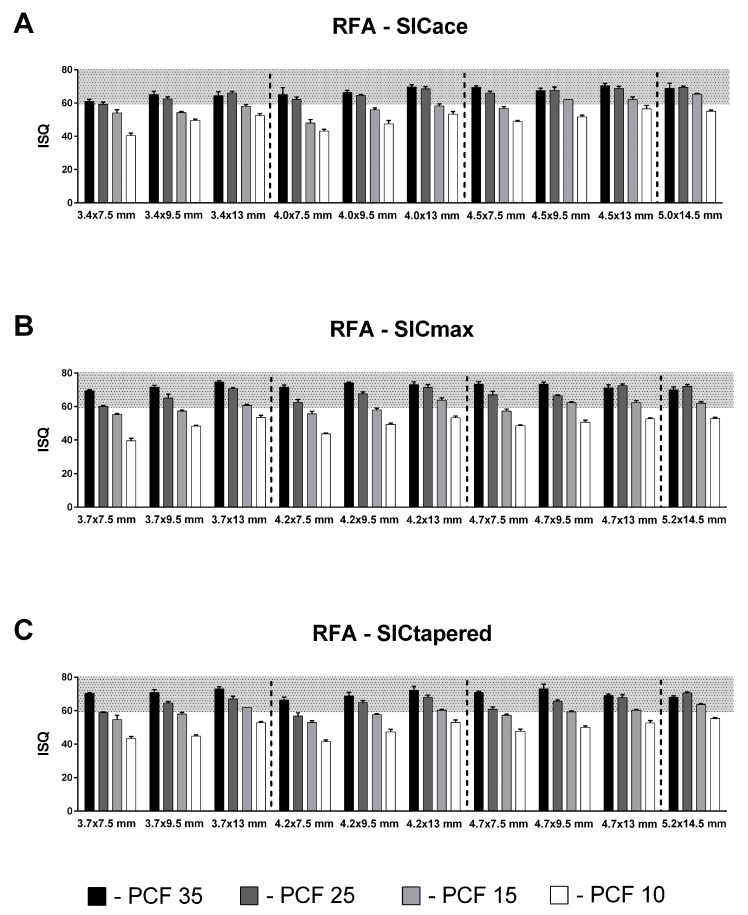
Resonance frequency analysis (RFA) of SICace (**A**), SICmax (**B**), and SICtapered (**C**) implants with different lengths and diameters in four different polyurethane foam block densities (35, 25, 15, and 10 pounds per cubic (PCF)). The grey shaded area represents the optimum range > 60 for ISQ values.

**Table 1 jfb-14-00469-t001:** Implant types, implant diameters (Ø), implant lengths, and standard drilling protocol used for the drilling study in solid rigid polyurethane foam blocks with four different densities. n = 6 per group.

Implant Type	Implant Ø (mm)	Implant Length (mm)	Last Drill Ø (mm)	Crestal Drill	Drilling Length
SICace^®^	3.4	7.5	3.1 (blue)	✔	Full (according to implant length)
SICmax^®^	3.7	9.5			
SICtapered^®^	3.7	13			
SICace^®^	4.0	7.5	3.25 (red)	✔	Full (according to implant length)
SICmax^®^	4.2	9.5			
SICtapered^®^	4.2	13			
SICace^®^	4.5	7.5	3.75 (yellow)	✔	Full (according to implant length)
SICmax^®^	4.7	9.5			
SICtapered^®^	4.7	13			
SICace^®^	5.0	14.5	4.25 (green)	✔	Full (according to implant length)
SICmax^®^	5.2				
SICtapered^®^	5.2				

**Table 2 jfb-14-00469-t002:** Contact surface area to bone in mm^2^ of implants. Ø: implant diameter; L: implant length.

Implant	Contact Surface Area to Bone (mm^2^)
SICace^®^ Ø 3.4 mm/L 7.5 mm	103.57
SICace^®^ Ø 3.4 mm/L 9.5 mm	130.24
SICace^®^ Ø 3.4 mm/L 13.0 mm	180.27
SICace^®^ Ø 4.0 mm/L 7.5 mm	123.45
SICace^®^ Ø 4.0 mm/L 9.5 mm	162.25
SICace^®^ Ø 4.0 mm/L 13.0 mm	218.32
SICace^®^ Ø 4.5 mm/L 7.5 mm	141.72
SICace^®^ Ø 4.5 mm/L 9.5 mm	186.64
SICace^®^ Ø 4.5 mm/L 13.0 mm	251.34
SICace^®^ Ø 5.0 mm/L 14.5 mm	333.53
SICmax^®^ Ø 3.7 mm/L 7.5 mm	106.57
SICmax^®^ Ø 3.7 mm/L 9.5 mm	137.36
SICmax^®^ Ø 3.7 mm/L 13.0 mm	198.26
SICmax^®^ Ø 4.2 mm/L 7.5 mm	120.02
SICmax^®^ Ø 4.2 mm/L 9.5 mm	162.71
SICmax^®^ Ø 4.2 mm/L 13.0 mm	217.26
SICmax^®^ Ø 4.7 mm/L 7.5 mm	143.17
SICmax^®^ Ø 4.7 mm/L 9.5 mm	191.82
SICmax^®^ Ø 4.7 mm/L 13.0 mm	277.13
SICmax^®^ Ø 5.2 mm/L 14.5 mm	345.59
SICtapered^®^ Ø 3.7 mm/L 7.5 mm	119.45
SICtapered^®^ Ø 3.7 mm/L 9.5 mm	139.11
SICtapered^®^ Ø 3.7 mm/L 13.0 mm	218.21
SICtapered^®^ Ø 4.2 mm/L 7.5 mm	110.28
SICtapered^®^ Ø 4.2 mm/L 9.5 mm	139.52
SICtapered^®^ Ø 4.2 mm/L 13.0 mm	213.71
SICtapered^®^ Ø 4.7 mm/L 7.5 mm	138.47
SICtapered^®^ Ø 4.7 mm/L 9.5 mm	176.32
SICtapered^®^ Ø 4.7 mm/L 13.0 mm	265.83
SICtapered^®^ Ø 5.2 mm/L 14.5 mm	329.49

**Table 3 jfb-14-00469-t003:** Differences in regression coefficients (*b*) for implant length and implant diameter between the block densities for SICace^®^, SICmax^®^, and SICtapered^®^. *b* illustrated the mean percentage change in insertion torque per 1 mm change in implant length or diameter. Significant differences in the regression coefficients (*b*) in PCF 10 are denoted by * (*p* < 0.05) or ^§^ (*p* < 0.1). The term ratio reflects the quotient of *b*_diameter_/*b*_length_.

		PCF 35			PCF 25			PCF 15			PCF 10		
		*b* (%)	CI95	CI95	*b* (%)	CI95	CI95	*b* (%)	CI95	CI95	*b* (%)	CI95	CI95
			(Low)	(High)		(Low)	(High)		(Low)	(High)		(Low)	(High)
SICace^®^	*Length*	12.3 *	10.7	13.8	7.8	5.5	10.1	3.2 ^§^	1.5	4.8	7.4	4.6	10.2
	*Diameter*	40.3 *	25.5	56.9	143.3 ^§^	122.3	166.3	81.7 ^§^	69.5	94.7	109.2	80.8	142.1
	*Ratio*	3.2			18.3			25.5			14.7		
SICmax^®^	*Length*	19.4 *	16.9	22	15.7 *	13.5	17.9	9.3 *	7.2	11.5	6.6	5.3	8
	*Diameter*	40.8 *	10.5	79.2	86.0	66.7	107.6	103.1	81.5	127.3	83.2	70.2	97.1
	*Ratio*	2.1			5.4			11.0			12.6		
SICtapered^®^	*Length*	16.4 *	14.4	18.4	15.0 *	12.6	17.6	7.5 *	5.8	9.2	4.9	3.9	5.8
	*Diameter*	62.5 ^§^	51.4	74.3	76.3 *	53.7	102.3	100.5 *	89.5	112.2	51.0	44.0	58.5
	*Ratio*	3.8			5.0			13.4			10.4		

## Data Availability

Not applicable.

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
