# Peer review of "Influence of Implant Macro-Design, -Length, and -Diameter on Primary Implant Stability Depending on Different Bone Qualities Using Standard Drilling Protocols—An In Vitro Analysis"

_jfb, 2023, doi:10.3390/jfb14090469_

Round 1

Reviewer 1 Report

The manuscript "Influence of implant macro design, -length, and -diameter on primary implant stability depending on different bone qualities using standard drilling protocols - an in-vitro analysis" reports the results of an experimental evaluation for the mechanical stability of dental implant in the absence of osseointegration using a polyurethane block model. This study shows a reasonable relation between implant length, diameter, and substrate density. The influence of implant shape is also shown, confirming the validity of the previously reported results. However, the current manuscript is not suitable for publication in Journal of Functional Biomaterials.

Comment #1 

The introduction is not adequate. The authors mention several controversies regarding the currently employed primary stability assessments, namely insertion torque (IT) and resonance frequency analysis (RFA). This controversy is not a problem that is resolved by the authors' study; rather, it corresponds to a limitation of the authors' study. Despite mentioning such issues, the study simply uses these methods without modifying or validating them. According to the authors' description, the purpose of this study is to develop recommendations for implant selection based on bone density and bone quality. This development is not guided by biological assessments such as marginal bone resorption, but only by IT and RFA. In other words, the authors should be in a position to explain the effectiveness of IT and RFA in this paper, but the authors' stance is written unclearly in the present manuscript. The authors are strongly suggested identifying the issue of this study.

Comment #2

The literatures are not correctly reviewed and misleadingly arranged. For example,

Page 2 (Line 61)

"However, Campos et al. argued that a lower insertion torque might promote more favorable bone repair around the inserted implants [26]."

Ref. 26 does not show favorable bone repair around the implant with lower insertion torque.

Page 2 (Line 63)

"In a systematic review and meta-analysis, Benic et al. recommended insertion torques ranging from ≥20 to 45 Ncm for immediate loading [27]."

Ref. 27 does not recommend applying insertion torques ranging from 20 to 45 Ncm for immediate loading.

Page 2 (Line 65)

"Conversely, increasing the insertion torque beyond a certain level may not be suitable for every implant system or bone type and could lead to over-compression of the bone [28]."

Ref. 28 does not state that insertion torques beyond a certain level may not be suitable or cause over-compression.

Page 2 (Line 82)

"Regarding implant geometry, both parallel-walled and tapered implants are available. Studies comparing the two geometries have provided contrasting results [40-42]. Tapered implants are considered more stable in low-density areas and minimize stress on the surrounding bone, but they may have a lower success rate and greater marginal bone loss after one year compared to parallel-walled implants [43-46]."

Refs. 40-42 are not contrasting. Ref. 40 shows the relation between insertion torque and micromotion, and is not an investigation on the implant shape. Both refs. 41 and 42 basically show higher insertion torques of tapered implants compared with that of cylindrical ones, and are not contrasting. Refs. 43-46 should not be simultaneously cited because they addressed different issues. Ref. 43 shows a positive clinical outcome of a tapered implant. Ref. 44 shows an in vitro mechanical study using a polyethylene block. Ref. 45 shows a better primary stability of a tapered implant through a clinical study. Ref. 46 is an investigation on marginal bone resorption, and no clear conclusions on implant shape was drawn. Therefore, Refs. 43-46 do not provide any negative evidences of tapered implants. In total, Refs. 40-46 are not properly reviewed.

The authors are strongly suggested reconsidering the way of reviewing previous works.

Comment #3

The criteria for evaluating primary stability of implants are declared without any validation. 

Page 5 (Line 182)

"The optimal range for IT was set from 25 to 50 Ncm and from 60 to 80 for ISQ values. Critical ranges were defined for IT values below 25 and above 50 Ncm and for ISQ values below 60."

No citation or validation was provided to define the optimal range and critical range. In the conclusion section, the authors recommend carefully monitoring the insertion torque when applying to hard bones based on this critical value.

Page 11 (Line 442)

"- The development of the insertion torque must also be observed in the hard bone in order to avoid pressure over-loads and necrosis in the peri-implant bone.

 Even with smaller implant sizes and standard drilling protocol, critical values (e.g., >50 Ncm) can be exceeded."

This note is only valid if the critical values are verified. The authors are strongly recommended making validation for the critical values of insertion toques and resonance frequencies.

Comment #4

The novelty of this study is questioned. The influence of macroscopic implant design on the insertion torque and the resonance frequency has been already investigated by one of the cited paper ref. 62. The influence of substrate density on the insertion torque and the resonance frequency has also been investigated by one of the cited paper ref. 42. 

Comment #5

The hypotheses tested in this study are considered trivial in the reviewers' view. The authors state their hypotheses as follows:

Page 2 (Line 95)

"The null hypothesis assumes no differences between the tapered and parallel-walled implants, as well as between implant diameter and length."

It is well known that the insertion torque increases with the implant length, implant diameter, or substrate density, due to the increased contact area. Although one of the cited paper ref. 44 shows higher torques for inserting tapered implants, this paper's authors reasonably attributed this result to the difference in microscopic design (thread geometry, thread pitch) of implants affecting the surface area rather than the macroscopic shape (tapered or parallel-walled). Thus, the reviewer recommends providing microscopic architecture of implants used in this study and discussing whether the comparison in this investigation was fair.

Reviewer 2 Report

The paper studied the relationship between bone implant device sizes and primary stability of the device right after the implantation. This is a niche issue but should be studied.  

I would like to invite the authors to pay attention to the following questions in the revision:

1) in introduction, please focus on what the current understanding is in terms of primary stability and dimensions of device sizes

2) it is not clear how torque is measured/controlled. how accurate and precise is the torque used in the study?

3) Statistical analysis section, lines 179 to 196, needs to be re-written to include information on what statistical analysis was done and sample sizes. Note this section is very important since the whole study is dependent on the statistical analysis. 

4) what is PCF? pounds per cubic??? (missing the unit)

5) there are several locations in the manuscript the authors reported results similar to this: F=38.9, df=3, p=0.000. what is this? what stats was done? why p is 0.000!?, is it even possible? what is df? 

6) in Table 2, "regression coefficient b" what is this? 

7) conclusions section needs to be re-written. This is not a powerpoint presentation and it should not be in bullet point format. 

no concerns

Reviewer 3 Report

Introduction

- to provide a complete state of the art on immediate loading, add few sentences describing how it led to the possibility of performing full arch rehabilitation treating full edentulous patients in few days. For this propose discuss and cite the following recent published article

Pera F, Pesce P, Menini M, Fanelli F, Kim BC, Zhurakivska K, Mayer Y, Isola G, Cianciotta G, Crupi A, Ambrogio G, Scotti N, Carossa M. Immediate loading full-arch rehabilitation using transmucosal tissue-level implants with different variables associated: a one-year observational study. Minerva Dent Oral Sci. 2023 May 16. doi: 10.23736/S2724-6329.23.04782-4.

Materials and methods

- Remove '' in the present study'' or ''in the present investigation'' throughout the text. They sound like repetition and it is already obvious.

- I had some difficulties understanding the sample size of the study. It is stated only later as ''This matrix forms the basis for the drilling axes of the planned 60 implant cavities per polyurethane foam  block and implant type (N = 6; a total of 720 implant site preparations for all four bone  qualities and implant types). '' I still don't really understanding it as it is confusing. Therefore, Cleary state at the beginning of the M&M section the sample size. After that, cleary report after each type of implant and after each variable considered the N= X through the text.

- To simplify the previous point, consider reporting a flow chart of the study at the beginning of the M&M section.It would help to immediately understand the study.

- Since the drillings were performed by a machine, report the speed of the drilling to guarantee the reproducibility of the study.

- Table 1, also add a column to report the N = X for each sub group, or if all the same x, report it in the table caption. 

Discussion

- Discuss if the null hypotheses were accepted or rejected based on the results of the study

- Discuss how an other possible in vivo consideration linked to the primary implant stability may be correlated on performing delayed loading following one or two stage approach.

Round 2

Reviewer 1 Report

The authors adequately answered to the reviewer's comments with substantial revisions. The reviewer believes that the quality of current manuscript is mostly acceptable for publication in Journal of Functional Biomaterials. The authors may consider the following minor comments.

Minor comments:

For the various implant bodies used in this study, the authors examined whether the primary stability is explained solely by the contact surface area specified by the manufacturer, or by macroscopic parameters such as the diameter and height. However, since the contact surface area values used in the analyses are not explained in the manuscript, it is difficult to understand the actual situation of the evaluation. The authors are recommended stating the contact surface area values used for the analyses or the method used to obtain them.

Furthermore, this analysis is based solely on insertion torque and not on resonance frequency. This is reasonabloe since the resonant frequency is supposed to be a measure of the average bond strength between the implant and bone. However, the authors state "The final null hypothesis assumes that the primary implant stability is solely determined by the contact surface area with the bone, irrespective of the implant's macro geometry and dimensions.". This is better to be replaced to "The final null hypothesis assumes that the insertion torque is ...".

Author Response

Reviewer 1

The authors adequately answered to the reviewer's comments with substantial revisions. The reviewer believes that the quality of current manuscript is mostly acceptable for publication in Journal of Functional Biomaterials. The authors may consider the following minor comments.

Minor comments:

For the various implant bodies used in this study, the authors examined whether the primary stability is explained solely by the contact surface area specified by the manufacturer, or by macroscopic parameters such as the diameter and height. However, since the contact surface area values used in the analyses are not explained in the manuscript, it is difficult to understand the actual situation of the evaluation. The authors are recommended stating the contact surface area values used for the analyses or the method used to obtain them.

We would like to thank the reviewer again for his time and effort. The comments and suggestions have significantly improved the manuscript.

We have added the manufacturer's specified contact area values of all implants to the manuscript in a new Table 2.

Furthermore, this analysis is based solely on insertion torque and not on resonance frequency. This is reasonabloe since the resonant frequency is supposed to be a measure of the average bond strength between the implant and bone. However, the authors state "The final null hypothesis assumes that the primary implant stability is solely determined by the contact surface area with the bone, irrespective of the implant's macro geometry and dimensions.". This is better to be replaced to "The final null hypothesis assumes that the insertion torque is ...".

We agree with the reviewer and changed the text as recommended.

Reviewer 3 Report

Thank you for addressing my points.

Author Response

Reviewer 3

Thank you for addressing my points.

We would like to thank again the reviewer for his comments and suggestions that improved the manuscript.